

# Daily GRACE gravity field solutions track major flood events in the Ganges-Brahmaputra Delta

Ben T. Gouweleeuw[1], Andreas Kvas[2], Christian Grüber[1], Animesh K. Gain[1], Thorsten Mayer-Gürr[2], Frank Flechtner[1], Andreas Güntner[1]

[1]Deutsches GeoForschungsZentrum GFZ, Telegrafenberg, 14473 Potsdam, Germany
[2]Graz University of Technology, Institute of Geodesy, 8010 Graz, Austria

*Correspondence to*: Ben T. Gouweleeuw (ben.gouweleeuw@gfz-potsdam.de)

**Abstract.** Two daily gravity field solutions based on observations from the Gravity Recovery and Climate Experiment (GRACE) satellite mission are evaluated against daily river runoff data for major flood events in the Ganges-Brahmaputra Delta (GBD) in 2004 and 2007. The trends over periods of a few days of the daily GRACE data reflect temporal variations in daily river runoff during major flood events. This is especially true for the larger flood in 2007, which featured two distinct periods of critical flood level exceedance in the Brahmaputra River. This first hydrological evaluation of daily GRACE gravity field solutions based on a Kalman filter approach confirms their potential for gravity-based large-scale flood monitoring. This particularly applies to short-lived, high-volume floods, as they occur in the GBD with a 4-5 year return period. The release of daily GRACE gravity field solutions in near real-time may enable flood monitoring for large events.

## 1 Introduction

Floods are dynamic events, which may only take hours to days to develop and drain. For monitoring purposes, Earth Observation products need to be available sufficiently frequently to capture the progressing stages of a flood event. Flood early-warning and forecasting systems additionally require information in near real-time to estimate probabilistic flood risk with typical lead times of a few days for larger river basins. Total water storage anomalies (TWSA) derived from temporal variations of the Earth's gravity field as observed by the Gravity Recovery and Climate Experiment (GRACE) twin-satellite mission (Tapley et al., 2004) have been shown to be a unique descriptor of large-scale flood events (Chen et al., 2010; Steckler et al., 2010). However, partly due to its coarse temporal (weekly to monthly) and spatial (> 150.000 km$^2$) resolution, the evaluation of the integrated information from GRACE on total water storage variations (i.e., variations of all surface and subsurface water stores) for flood monitoring or forecasting has been limited.

Reager and Famiglietti (2009) proposed a regional monthly flood potential index using a GRACE-based saturation deficit approach, which was quantitatively evaluated by Molodtsova et al. (2016) for the continental United States. While the index agreed well with observed floods on regional and even local scales, an increased forecasting skill was found for large-scale, long duration floods during summer. Reager et al. (2014) established a relationship between gauged river flow and GRACE-derived basin-wide water storage for the March-June 2011 Missouri river flood, potentially increasing forecasting lead time to several months. The same 500-year flood event was evaluated by Reager et al. (2015), who assimilated monthly GRACE-





derived TWSA into a land surface model, which enables state disaggregation of the vertically-integrated TWSA, a downscaling of GRACE's coarse spatial resolution and near real-time analysis beyond the latest GRACE data release.

Currently, the latency in data product processing and release consists of a nominal time delay of the GRACE Level-1 instrument data (11 days) and of the derived monthly global Level-2 gravity field products (60 days). Temporal sampling is at best 7-10 days, but most reliably one month, caused by a need to accumulate GRACE observations over this time period. Both latency and temporal averaging currently limit the potential use of the GRACE Level-3 products (i.e., TWSA) for flood monitoring and early-warning systems. Daily GRACE gravity field solutions based on a Kalman filter approach (Kurtenbach et al., 2012) have been validated against ocean signals in the Antarctic circumpolar current, which showed the possibility to detect high-frequent ocean mass variations (Bergman and Dobslaw, 2012). Recently, Sakumura et al. (2016) proposed a method for improved high-frequency signal capture via a regularized sliding window mascon (RSWM) product, which approximates a daily GRACE solution using a moving weighting scheme of 21 adjacent days of observational data, with about 10 days delay.

For the first time, this study presents a hydrological evaluation of daily GRACE gravity field solutions based on a Kalman filter approach, which are scheduled for an operational run in 2017 with a time delay of just 5 days, by comparing the time series to observed river runoff in the Ganges-Brahmaputra Delta (GBD) under flood conditions. The world's largest river delta, situated at the confluence of two river systems with a combined discharge surpassed only by the Amazon and the Congo, is subject to short-lived flooding throughout summer and early autumn each year. In a typical year, 20-30% of Bangladesh, which occupies most of the GBD, can be inundated for days during the monsoon, mostly in low-lying fields (Steckler et al., 2010). Major flooding events occur with a return period of 4-5 year (Hopson and Webster, 2010), causing widespread devastation in this densely populated part of the world. Two of these major flood events coincide with the GRACE mission's lifetime (March 2002 – present). In July 2004, the Brahmaputra exceeded critical flood levels twice in a fortnight inundating 38% of the country (Best et al., 2007). In July and September 2007, two separate major flood peaks in the Brahmaputra caused inundation lasting weeks affecting 42% of the country (Islam et al., 2010).

## 2 GRACE data processing

Compared to monthly solutions, the limited spatial coverage within one day does not allow for GRACE to observe the full gravity field signal alone. Limited spatial sampling in East-West direction means that GRACE contributes little to no information to potential coefficients with orders higher than approximately 15. It is therefore necessary to introduce additional information to obtain reliable estimates of the full global gravity field signal. Applied to the determination of daily gravity field variations, this means that information on how the variable gravity field evolves with time is required. Since geophysical processes are not random, one can assume that the Earth's time-variable gravity field does not change arbitrarily from one day to the next. A simplified model of this assumption is a first-order Markov process:



$$x_t = B\, x_{t-1} + w, \quad w \sim N(0, Q). \tag{1}$$

Here, the matrix B represents a linear predictor from epoch t-1 to epoch t. The accuracy of the prediction is characterized by the normally distributed noise vector w and its covariance matrix Q. As stated in Moritz (1980), B and Q can be estimated by:

$$x_t = \Sigma_\Delta \Sigma^{-1} x_{t-1} + w, \quad w \sim N(0, \Sigma - \Sigma_\Delta \Sigma^{-1} \Sigma_\Delta^T), \tag{2}$$

where $\Sigma$ and $\Sigma_\Delta$ are the auto- and cross covariance matrices of the process. In practice however, the true covariance matrices in Eq. 2 are unknown and have to be derived empirically. Applied to GRACE, the process to be modelled is the residual gravity field signal that is present in the observations after other effects, such as long-term, secular, as well as non-tidal ocean and atmosphere variations have been reduced. The main geophysical constituents left in the GRACE data are therefore continental hydrology, cryosphere, solid earth and errors in the background models (Kurtenbach et al., 2012).

For the daily solutions of the ITSG-Grace2014 release (Mayer-Gürr et al., 2014), the model output of the updated ESA Earth System Model (Dobslaw et al., 2015) is used to approximate the unknown covariance structure of this residual gravity field signal. The 6-hourly model output is resampled to one day using daily averaging. These daily averages are subsequently reduced by their sample mean, trend and annual signal. The resulting state vectors $x_t$ are then used to approximate the covariance matrices using the estimators

$$\Sigma \approx \bar{\Sigma} = \frac{1}{N} \sum_{k=1}^{N} x_t x_t^T \tag{3}$$
and
$$\Sigma_\Delta \approx \bar{\Sigma}_\Delta = \frac{1}{N-1} \sum_{k=2}^{N} x_{t-1} x_t^T, \tag{4}$$

respectively.

For the daily GRACE solutions from GFZ, empirical functions for the auto- and cross-covariance matrices are derived from harmonic analysis of WaterGAP Global Hydrology Model (Döll et al., 2003) output, the Atmosphere and Ocean De-aliasing Level-1B (AOD1B) products (Dobslaw et al., 2013), as well as GFZ monthly solutions (Dahle et al., 2012). The GFZ daily solution employs radial basis functions (RBF) to map the measured GRACE twin-satellite range-accelerations into the mass anomaly (or water equivalent) surface layer. Precise dynamic orbits of the satellite trajectories using Global Positioning System (GPS) data complement the micro-wave measurements of acceleration in-line-of-sight between the GRACE twin satellites (Gruber et al., 2014).

A widely used tool to combine prior information of the underlying process and measurements of this process is the Kalman filter (Kalman, 1960). Given the previously derived process model, one can predict the Earth's gravity field for the following epoch and compute the corresponding accuracy information using covariance propagation. This prediction is consequently updated by forming a weighted mean between the predicted and observed state is formed, where



the weights of both observation groups are determined by their respective covariance matrix. For the ITSG-Grace2014 daily solutions (Kurtenbach et al., 2012), the Kalman filter is run in forward as well as backward direction. This allows for a smoothed estimate of the state vectors by computing the weighted minimum-variance mean of both time series (Rauch et al., 1965). Since the future purpose of the daily GRACE solutions is a near real-time (NRT) service mode, the GFZ RBF solutions

in this study have been processed with the Kalman filter in forward mode only.

To derive gridded TWSA from the ITSG-Grace2014 gravity field potential coefficients, the processing scheme used for the GRACE land water mass grids provided by GRACE Tellus is followed (Swenson, 2012; Landerer and Swenson, 2012). Since the Earth's oblateness (C20 coefficient) determined by GRACE is subject to a large uncertainty, it is replaced by the solution from a satellite laser ranging (SLR) time series (Cheng et al., 2011). The GFZ RBF solutions also make use of the order 1 and

2 coefficient of these SLR series. The transformation from the center-of-mass to center-of-figure frame is realized using the degree one coefficients estimated using the method described in Swenson et al. (2008). Glacial isostatic adjustment correction has been applied using the model from Geruo et al. (2013). For comparison, also the monthly solutions of ITSG-Grace2014 (Mayer Gürr et al., 2014) and GFZ RL05a (Dahle et al., 2012) are considered. An anisotropic filter DDK2 (Kusche et al., 2009) is applied to the unconstrained monthly gravity field solutions, while for the Kalman solutions no spatial filtering is

necessary due to the regularization in the estimation process. Both monthly and daily gravity field solutions are then propagated to TWSA on a $1^0$ x $1^0$ grid (~100 km at $25^0$ latitude). The actual spatial resolution of the gridded TWSA, however, is lower with approximately 330 km and 500 km for monthly and daily solution, respectively.

## 3 Results

### 3.1 Process dynamics

To illustrate the possible impact of the process dynamics that may originate from the hydrological models on the daily Kalman GRACE solutions, three different daily gravity time series are computed using stochastic information from three different models, i.e. the WaterGAP Global Hydrological Model (WGHM), the Land Surface Discharge Model (LSDM) and the Global Land Data Assimilation System (GLDAS), respectively. The GRACE input is identical for each time series. It can therefore be assumed that differences in the gravity field time series are caused by the different predictors only. Figures 1a-c show the

water storage variability and the differences between the three hydrological/land surface models and the three daily GRACE solutions computed using the stochastic information of these models, as well as the differences between the GRACE solutions and the hydrological models. For each time series, secular and annual variations are removed. The results show that, first, the differences between the three hydrological models may vary as much as the variability of the daily signal simulated by each hydrological model alone (Figure 1a). Secondly, the difference between the GRACE solutions (Figure 1b) is consistent and

relatively low, while the variability of each GRACE solution alone is consistent and relatively high. Finally, the difference between a GRACE solution and a particular hydrological model varies for each hydrological model, but is consistent for any



of the three GRACE solutions (Figure 1c). It is therefore concluded that there is hardly any model-specific information left in the GRACE solutions and that the information from the actual GRACE observation clearly dominates.

## 3.2 The 2004 flood

Figures 2 and 3 show a succession of daily GRACE TWSA for ITSG-Grace2014 on a) 13 July, b) 21 July, and c) 27 July 2004

and for GFZ RBF on a) 11 July, b) 24 July, and c) 31 July 2004, respectively, together with d) the monthly ITSG-Grace2014 and GFZ RL05a gravity field solution for July 2004, respectively. Elevated daily TWSA values for the larger GBD (~210.000 km² area, including Bangladesh and parts of North-East India) on these dates show a progression of widespread flooding in the delta within half a month (a) just before, (b) during and (c) after peak flooding, which cannot be resolved temporally in (d) the monthly solutions. Due to the different filtering techniques (see above), the signal amplitude of the daily and monthly

gravity field solutions also cannot be compared directly. For example, lower TWSA in the monthly solutions may be partly due to post-processing (filtering) of the GRACE observations resulting in signal attenuation (Kusche et al., 2009). Different process dynamics, which are also partly reflected in different noise levels cause the daily solutions to peak on different days. Spatially, the GFZ RBF solution shows more variation between the three daily snapshots, while high TWSA values are slightly more concentrated. All solutions, however, show comparable spatial patterns of increased TWSA along the Brahmaputra

North-East of the GBD. Flood stages were reached between 10 and 26 July 2004 in the Brahmaputra at Bahadurabad (B) river runoff station, while the Ganges at Hardinge Bridge (H) did not reach critical flood levels (Hopson and Webster, 2010).

Figure 4 shows time series of area mean values of daily and monthly GRACE TWSA and observed daily river runoff anomalies, together with daily precipitation surplus. The added value of the daily GRACE solutions is demonstrated by means of a high-pass filter (HPF) with a near-monthly (31-day) window (upper panel). The HPF is applied to daily anomalies (mean

reduced) of total water storage, river runoff and precipitation for the period between 2003 (start of the record of daily GRACE solutions) and 2009 (end of daily river runoff record). The remaining part of the daily time series, the low-pass component of the filter (LPF), is also shown (lower panel), together with a series of cubic spline interpolations (both panels, dotted line) fitted to the monthly GRACE gravity field solutions (both panels, step line).The LPF and monthly time series have the mean seasonal cycle (2003-2009) removed in order to focus on the sub-seasonal anomalies. To facilitate comparison with area mean

TWSA, the observed daily river runoff at the two stations is combined. Daily precipitation is taken from the WFDEI ('WATCH Forcing Data methodology applied to ERA-Interim data') dataset (Weedon et al., 2014), using Global Precipitation Climatology Centre (GPCC) precipitation totals. Precipitation surplus, computed as an area mean for the GBD, represents the positive HPF signal (upper panel). The HPF river runoff signal peaks on 23 July, coinciding with the reported flood stage of the Brahmaputra at Bahadurabad station between 10 and 26 July (Hopson and Webster, 2010). The HPF water storage signals

of ITSG-Grace2014 and GFZ RBF peak 2 days earlier and 1 day later, on 21 July and 24 July, respectively. Correlation, expressed as Pearson's linear correlation coefficient (r), between the HPF signals of precipitation ('surplus') and river runoff anomalies (Table 1) is strongest during a period, in which the seasonally corrected LPF signals of the daily GRACE solutions





increase and peak, from mid-June to the end of July (r=0.62 at a 9 day time lag). This also applies to the HPF signals of river runoff and water storage anomalies (r=0.89 at a 2 day time lag and r=0.60 at a 1 day time lag for ITSG-Grace2014 and GFZ RBF, respectively). The short time lag in the maximum correlation of the HPF signals indicates that part of the river runoff does not enter longer-term floodplain or subsurface stores during this time, but is transported through the GBD and discharged

into the ocean. The strong correlation of the HPF signals of precipitation, runoff and storage during this period suggests river runoff and precipitation contribute to flooding not only when regional water storage peaks, but also shortly prior to that, when the available stores are still filling up. A successive drop in the level of regional water storage, manifested by a decrease of the LPF signal of the daily GRACE solutions, is reflected in a weaker correlation of the HPF signals of precipitation, runoff and storage. River runoff data is missing to evaluate the correlation for another period of increased regional water storage, which

coincides with a period of extreme precipitation surplus with a peak of 106 mm on 6 October.

### 3.3 The 2007 flood

The Brahmaputra exceeded critical flood levels at Bahadurabad during two distinct periods in 2007, i.e. 26 July-6 August and 8-17 September (Hopson and Webster, 2010). As in 2004, the Ganges at Hardinge Bridge did not reach critical flood levels. Figures 5 and 6 show a sequence of daily GRACE TWSA for ITSG-Grace2014 for the first flood peak on a) 24 July, b) 3

August, and c) 13 August 2007 and for GFZ RBF on a) 25 July, b) 3 August, and c) 11 August 2007, respectively, together with d) the monthly ITSG-Grace2014 and GFZ RL05a gravity field solutions for August 2007, respectively. Figures 7 and 8 show a sequence of daily GRACE TWSA for ITSG-Grace2014 for the second flood peak on a) 1 September, b) 12 September, and c) 26 September 2007 and for GFZ RBF on a) 2 September, b) 8 September, and c) 21 September 2007, respectively, together with d) the monthly ITSG-Grace2014 and GFZ RL05a gravity field solutions for September 2007, respectively.

Again, the series of daily snapshots reflect the progression of flooding in the delta (a) just before, (b) during and (c) after peak flooding, which is beyond the temporal resolution of (d) the monthly solution. As in 2004, the two daily solutions do not necessarily peak on the same day (second flood peak), while spatial patterns of peak TWSA show slightly different areas of concentration.

Figure 9 shows time series of the area mean values of daily and monthly GRACE TWSA and observed daily river runoff

anomalies, together with daily precipitation surplus. The high-pass filtered daily gravity field solutions clearly reflect the two peaks of the HPF daily river runoff anomalies on 31 July and 13 September, and a smaller peak in-between on 21 August.

As in 2004, correlation between the HPF signals of precipitation ('surplus') and river runoff anomalies (Figure 9, upper panel) is strongest during a period, in which the LPF signals of the daily GRACE solutions peak (r=0.61 at an 8 day time lag). In 2007, however, the period of strong correlation is defined by a flattened peak, during which water storage plateaus and which

lasts from the end of July to mid-October with only a small drop around 23 August. Similarly, the correlation between the HPF signals of river runoff and water storage anomalies peaks during this period (r=0.95 and r=0.77 at a 1 day time lag for ITSG-Grace2014 and GFZ RBF, respectively). This indicates that the available water storage in the area is close to capacity during an extended period of time (~2 months) and additional inputs of runoff and precipitation can only be stored in the river during



this time. The stronger correlation of anomalies of water storage with river runoff into the GBD than with precipitation within the GBD suggests that river runoff is a stronger driver for major flooding than precipitation in the GBD, particularly for the 2007 flood (Steckler et al., 2010; Islam et al., 2010]. Both in 2004 and 2007, water storage increases slightly before river runoff, possibly due to the large GRACE footprint, which detects increasing water storage in the upstream reaches of the

Brahmaputra and Ganges rivers earlier than recorded as runoff at the gauges (see e.g., Figure 2).

The strong correlation of the seasonally-corrected low-frequency signals (Figure 9, lower panel and Table 1) indicates that river runoff (volume of ~700 km$^3$ and ~770 km$^3$ for 2004 and 2007, respectively) also contributes predominantly to sub-seasonal storage variations (~100 km$^3$ and ~110 km$^3$ for 2004 and 2007, respectively), while correlation with precipitation in the GBD (~450 km$^3$ and ~475 km$^3$ for 2004 and 2007, respectively) is weaker. This corresponds with the results of the HPF

signals for 2004 and suggests that river runoff into the GBD and, to a lesser degree, precipitation within the GBD not only trigger major flooding when water stores in the region have already been filled and peak, but they also contribute to creating these conditions for flooding. Table 1 further shows that the correlation of the LPF signal of the daily GRACE solutions with runoff exceeds that of the monthly GRACE data, including the fitted series of cubic spline interpolations. The stronger correlation indicates that the seasonally-corrected LPF signal contains additional information at this frequency compared to

the DDK filtered monthly GRACE data. A similar result for the 2004 flood is reported by Sakumura et al. (2016), who found good agreement of the signal amplitude at (sub-)seasonal frequencies for their RSWM product with in situ data and the Center of Space Research (CSR) RL05 monthly solutions.

For both floods, the added value of the daily GRACE solutions is further illustrated in Figures 4 and 9. Although phase and amplitude differ, the dynamics of the series of cubic spline interpolations (CSI, dotted line) fitted to the monthly GRACE

solutions (step line) are comparable to that of the LPF signal of daily GRACE solutions and river runoff anomalies in 2004 (Figure 4, lower panel). In 2007, the CSI series resolve the flood as a single event, while the LPF signal of daily GRACE solutions is able to distinguish the two flood peaks (Figure 9, lower panel). Additionally, the HPF signal of the daily GRACE solutions (upper panels) is able to capture the HPF signal of the river runoff anomalies, particularly during the period, in which regional water storage is filling up and peaks (2004) or plateaus (2007).

This is further illustrated in Figure 10, which shows the frequency-separated anomalies of river runoff vs. total water storage in the GBD for the flood years of 2004 and 2007 (Riegger and Tourian, 2014; Reager et al., 2015; Sproles et al., 2015). Generally, the hypothesis that storage can drive river runoff tends to indicate a slower process evolution through subsurface water storage and baseflow generation, expressed by a strong correlation at longer (monthly) time scales. Daily HPF runoff is expected to be more variable than daily HPF storage, caused by precipitation than runs off quickly and doesn't enter storage

for a significant amount of time. The fact that there is still a strong correlation between daily HPF storage and HPF river runoff, particularly for the 2007 flood (Table 1), points towards a scenario, which sees the variation of daily total water storage reflecting the inflow of river water into the delta, which can only be stored in the river during the time of flooding, when the available water storage in the area is at (near) capacity. However, while trends of the HPF signal of daily water storage and river runoff anomalies show agreement over periods of a few days, the higher frequency content of the daily solutions is not





reflected in the daily river runoff. This high-frequency variation is attributed to process noise of the Kalman filter approach and a repeat period of 4 days for GRACE to pass over the GBD. Propagation of the full formal error matrix to the area mean value for each time step estimates the resulting noise level in the ITSG-Grace2014 and GFZ RBF daily solutions at approximately 1 cm TWSA and 1.45 cm TWSA, respectively. These numbers are confirmed by an empirical estimate using

the methodology of Bonin et al. (2012). The difference in apparent noise in the two daily time series can primarily be attributed to the process models employed, which result in different temporal constraints and degree of spatial filtering.

## 4 Conclusions

Daily GRACE gravity field solutions have been evaluated against daily river runoff data for major flood events in the Ganges-Brahmaputra Delta (GBD) in 2004 and 2007. Compared to the monthly gravity field solutions, the trends over periods of a

few days in the daily gravity field solutions are able to reflect temporal variations in river runoff during major flood events. This is especially true for the larger flood in 2007, for which three consecutive peaks in daily river runoff, of which two exceeded critical flood levels, are replicated. The daily temporal resolution is sufficiently high to reflect these area mean variations of water storage anomalies, but the spatial resolution is too low to accurately locate the flood affected area. GRACE total water storage anomaly data is an integrated observation of multiple water storage compartments (e.g., river

storage, lakes and floodplains, soil moisture, groundwater). The strong correlation of high-pass filtered daily river runoff anomalies and TWSA suggests that river water constitutes a large part of the daily total water storage variation during flooding, when other water stores are at (near) capacity and cannot absorb additional inputs of runoff or precipitation. Strong correlation further indicates that the remaining part of the daily GRACE data, the low-pass component of the filter, also contains additional information at this frequency compared to the monthly GRACE data. The analysis further suggests that river runoff into the

GBD, mainly generated by precipitation in the upstream reaches of the Ganges-Brahmaputra basin, is a stronger driver for major flooding than precipitation in the GBD, particularly for the greater flood of 2007. This first hydrological evaluation of the daily GRACE gravity field solutions based on a Kalman filter approach confirms their potential for gravity-based large-scale flood monitoring. This particularly applies to short-lived, high-volume floods, as they occur in the GBD with a 4-5 year return period. These results imply that with the release of the daily gravity field solutions in

near real-time, flood monitoring may be supported for large events.

## Acknowledgements and data

This research is funded by the European Union's Horizon 2020 project European Gravity Service for Improved Emergency Management (EGSIEM) under grant agreement No 637010. We thank Dr J.T. Reager for his comments and suggestions, which greatly helped to improve the manuscript. Discharge station data is kindly provided by the BangladeshWater and

Development Board, Flood Forecasting and Warning Centre, Dhaka, Bangladesh. Degree-1 estimates, C20 SLR time series




and GIA model have been acquired from http://grace.jpl.nasa.gov. The daily GRACE gravity solutions solutions are available from ftp://ftp.tugraz.at and ftp://egsiem@gfzop.gfz-potsdam.de.

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





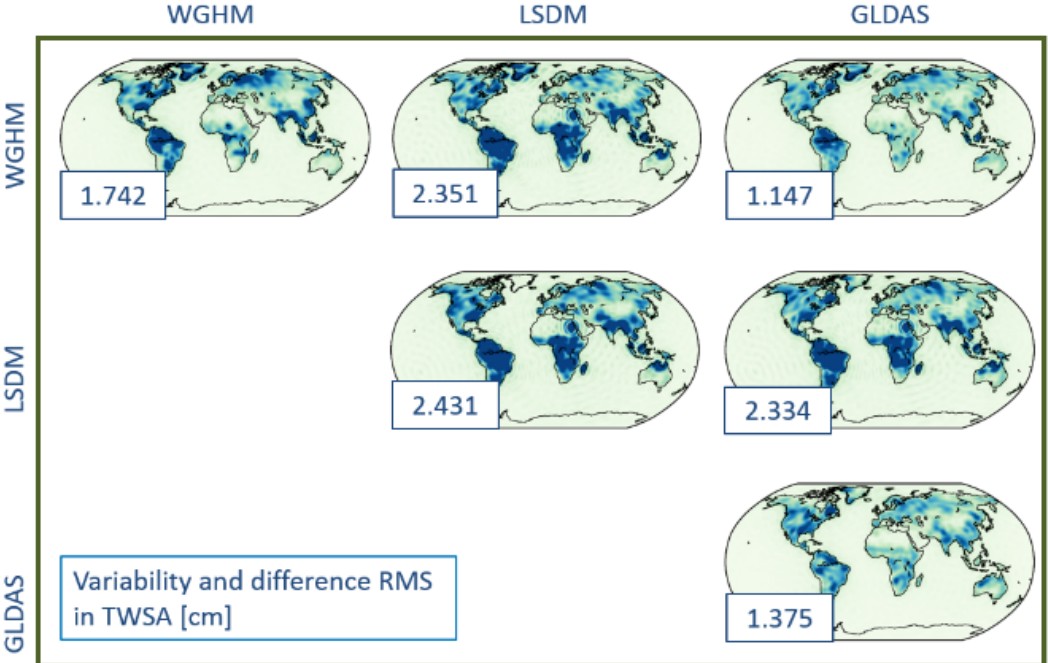

**Figure 1a. Global average daily water mass variability (main diagonal) and RMS differences (off diagonal) between the used hydrological models, expressed as TWSA (2003-2014, secular and annual variations removed).**

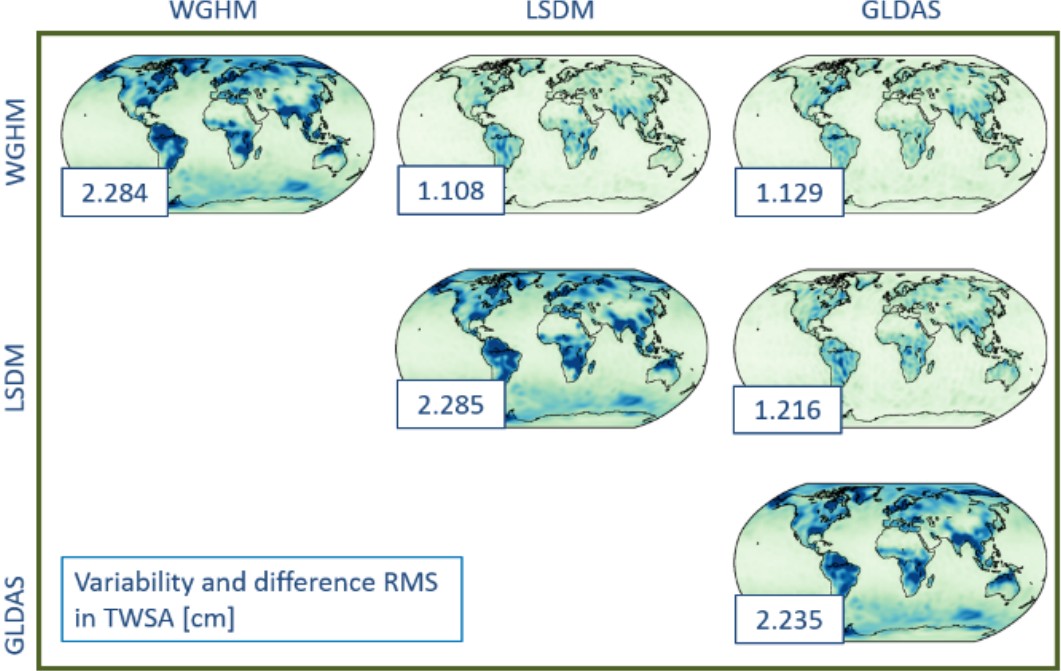

5    **Figure 1b. Global average daily variability (main diagonal) and RMS differences (off diagonal) of the derived daily GRACE solutions, expressed as TWSA (2003-2014, secular and annual variations removed).**





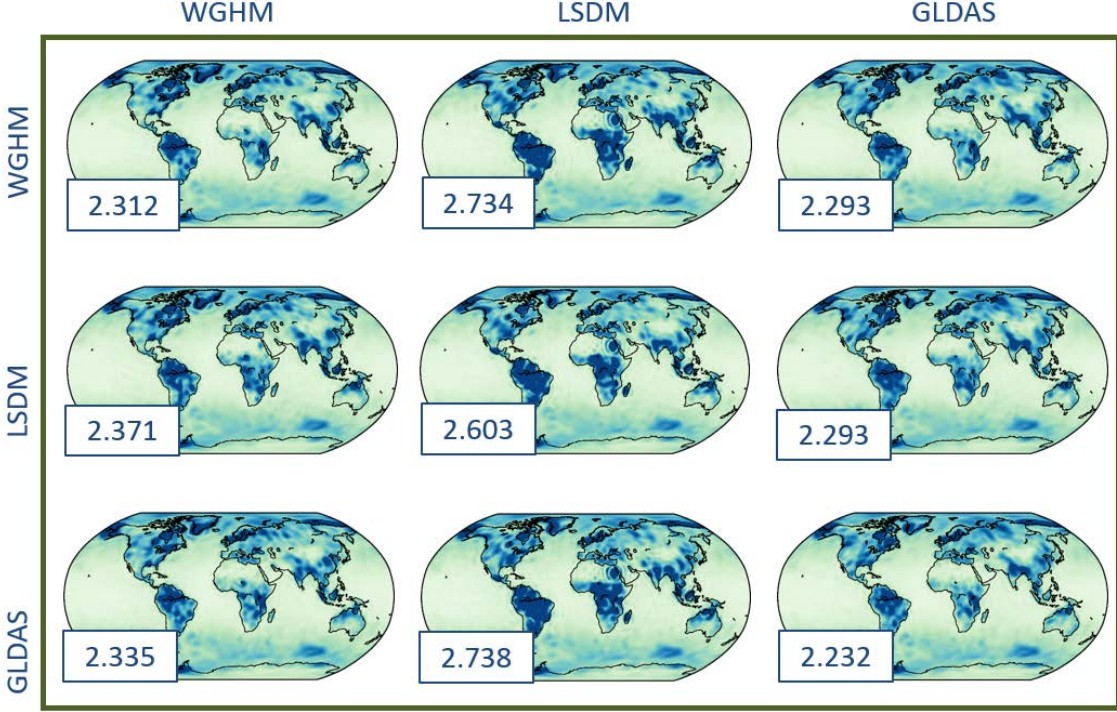

**Figure 1c. Global RMS differences in cm between the daily GRACE solutions (rows) and the used hydrological models (columns), expressed as TWSA (2003-2014, secular and annual variations removed).**





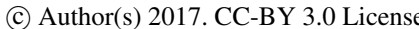

**Figure 2. Daily and monthly water storage anomalies of the ITSG-Grace2014 solution for the Ganges-Brahmaputra Delta (outline indicated) on (a) 13 July 2004 (b) 21 July 2004 (c) 27 July 2004 and (d) July 2004. Increased TWSA is related to flooding. River runoff stations are Bahadurabad (B) in the Brahmaputra and Hardinge Bridge (H) in the Ganges.**





**Figure 3. Daily and monthly water storage anomalies of the GFZ RBF solution for the Ganges-Brahmaputra Delta on (a) 11 July 2004 (b) 24 July 2004 (c) 31 July 2004 (d) July 2004. Key as in Figure 2.**





**Figure 4. Daily and monthly area mean anomalies of GRACE total water storage (TWSA) and daily river runoff (left axis) together with precipitation surplus (right axis) for the GBD in 2004. The daily data is separated in a high-frequency (upper panel) and a low-frequency signal (lower panel) by means of a 31-day high-pass filter. A series of cubic spline interpolations (dotted line) is fitted to the monthly solutions (step line). The squares represent (the number of) monthly values. The low-frequency signals and the monthly solutions have the mean seasonal cycle removed.**





**Figure 5. Daily and monthly water storage anomalies of the ITSG-Grace2014 solution for the Ganges-Brahmaputra Delta on (a) 24 July 2007 (b) 3 August 2007 (c) 13 August 2007 and (d) August 2007. Key as in Figure 2.**





**Figure 6. Daily and monthly water storage anomalies of the GFZ RBF solution for the Ganges-Brahmaputra Delta on (a) 25 July 2007 (b) 3 August 2007 (c) 11 August 2007 (d) August 2007. Key as in Figure 2.**




**Figure 7. Daily and monthly water storage anomalies of the ITSG-Grace2014 solution for the Ganges-Brahmaputra Delta on (a) 1 September 2007 (b) 12 September 2007 (c) 26 September 2007 and (d) September 2007. Key as in Figure 2.**





**Figure 8. Daily and monthly water storage anomalies of the GFZ RBF solution for the Ganges-Brahmaputra Delta on (a) 2 September 2007 (b) 8 September 2007 (c) 21 September 2007 (d) September 2007. Key as in Figure 2.**





**Figure 9. Daily and monthly area mean anomalies of GRACE total water storage (TWSA) and daily river runoff (left axis) together with precipitation surplus (right axis) for the GBD in 2007. Key as in Figure 4.**





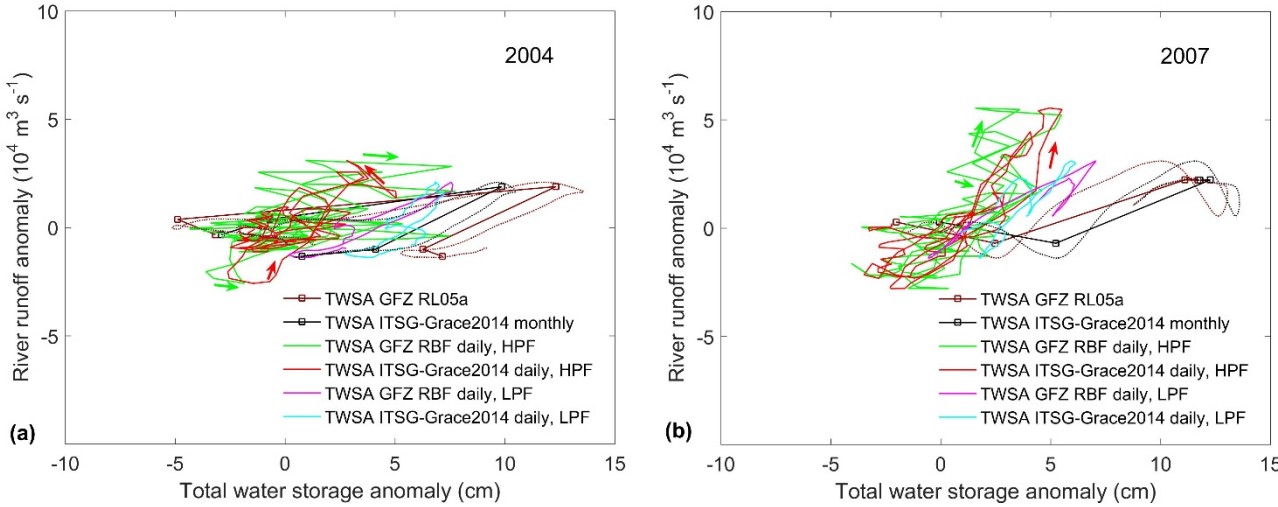

**Figure 10. Anomalies of river runoff vs. total water storage for a) 2004 and b) 2007. A high-pass filter (HPF) with a 31-day window separates daily data in a HPF and a low-pass frequency (LPF) component. A cubic spline interpolation (dotted line) is fitted to the monthly data. The LPF signals and monthly data have the mean seasonal cycle removed. Arrows indicate (local) directions in the HPF signal hysteresis loop.**





**Table 1. Correlation of daily and monthly TWSA and daily precipitation with river runoff anomalies in(to) the GBD for the years of major flooding in 2004 and 2007. The daily data is high-pass filtered (HPF) with a 31-day window. The low-pass filter (LPF) component of the filter, the monthly data and a series of cubic spline interpolations (CSI) fitted to the monthly data have the mean seasonal cycle removed. Pearson's linear correlation coefficient (r), sample size (n) and a (p-) value for testing the hypothesis of no correlation are indicated. Time lag, if applicable, is indicated in days between brackets (positive time lag denotes a delayed river runoff response).**

| | 31-day High Pass Filter | | | 31-day Low Pass Filter* | | |
|---|---|---|---|---|---|---|
| | | | | | | 2004 |
| | r | N | *p* | r | n | *p* |
| Precipitation | 0.41 (+8d) | 148 | < 0.01 | 0.64 (+7d) | 148 | < 0.01 |
| ITSG-Grace2014 daily | 0.69 (+1d) | 148 | < 0.01 | 0.95 (-1d) | 148 | < 0.01 |
| GFZ RBF daily | 0.42 (-2d) | 148 | < 0.01 | 0.79 | 148 | < 0.01 |
| ITSG-Grace2014 monthly* | | | | 0.59 | 5 | 0.29 |
| GFZ RL05a* | | | | 0.22 | 5 | 0.73 |
| ITSG-Grace2014 CSI* | | | | 0.70 (-6d) | 148 | < 0.01 |
| GFZ RL05a CSI* | | | | 0.33 (-7d) | 148 | < 0.01 |
| | | | | | | 2007 |
| | r | N | *p* | r | n | *p* |
| Precipitation | 0.43 (+9d) | 114 | < 0.01 | 0.77 (+12d) | 114 | < 0.01 |
| ITSG-Grace2014 daily | 0.87 (+1d) | 114 | < 0.01 | 0.95 (-1d) | 114 | < 0.01 |
| GFZ RBF daily | 0.70 (+1d) | 114 | < 0.01 | 0.85 | 114 | < 0.01 |
| ITSG-Grace2014 monthly* | | | | 0.79 | 4 | 0.21 |
| GFZ RL05a* | | | | 0.85 | 4 | 0.15 |
| ITSG-Grace2014 CSI* | | | | 0.72 (-1d) | 114 | < 0.01 |
| GFZ RL05a CSI* | | | | 0.74 (-2d) | 114 | < 0.01 |

*mean seasonal cycle removed

