# Peer review of "Daily GRACE gravity field solutions track major flood events in the Ganges-Brahmaputra Delta"

_Hydrology and Earth System Sciences, 2016_

## Short Comment (SC1) · 28 Mar 2017

Please note the 2nd Co-Author to go by the name of Gruber, i.e. without umlaut.

---

## Referee Comment (RC1) · Anonymous Referee #1 · 15 Aug 2017

Dear authors,

This is an interesting contribution to examine daily GRACE solutions that include much richer spectrum than monthly solutions. Visibly high correlation between the daily solutions and runoff data is promising. I would expect many scientific applications are plausible using the daily solutions. However, this manuscript needs some improvement for publications. Please find my comments below.

(P2,L14). Delivery of GRACE L1 takes 11 days as mentioned in introduction. How is it possible to produce the daily solution in 5 days? Do you have another improvement for the L1 data processing?

[Figure]

(P3,Eq(3-4)). Please explain explicitly about k and N.

(3.1 Process dynamics). I understood that two daily solutions differ mostly from realization of matrix B that is empirically estimated by models. And, Figure 1 shows that solutions are close to each other regardless of models for B. Why don't you present the variance and co-variance matrix first?

(P5,L4-9). Why the sampling dates of daily solutions are not the same? Is this due to noise so that some daily solutions could not present such nice snapshots? Indeed, GFZ RBF is corrupted more noise as shown in Figures 4 and 9.

(P5, L9 "Due to the different...."). However, monthly mean of daily solutions need to be compared with monthly solutions. This is important because you examined the LPF of daily solutions. In terms of month-to-month variation, monthly solutions should be superior to monthly mean of daily solutions.

(P5, L30) Correlation coefficients, r, in Table 1 are based on entire flooding years (am I right?). But you talked about r during flooding epochs in the text. I was quite confused during reading this part. So, please be clear this. It seems that many r in the table is not meaningful in the text. You may include r during flooding events in the table. Furthermore, r during flooding events are statistically significant? What are p values?

(The 2007 flood) Similar comments for 'the 2004 flood' are also applied here.

(P7, L18) "..added value of the daily GRACE..") Did you mean ".. the monthly GRACE.." ?

(Figure 10). Explanation for Figure 10 is quite terse. More explanations would be necessary for GRACE audience.

(Figures 4 and 9) Both LPF time series quite differ from each other. This is not consistent with results in Figure 1.
* * *
653, 2017.

---

## Referee Comment (RC2) · N. Tangdamrongsub (Referee) · 16 Oct 2017

The paper utilizes GRACE daily solution on detecting major flood events over the Ganges-Brahmaputra basin. Results from two different daily solutions are investigated and compared with the monthly solution from the same data centers. The paper shows that the daily solutions are capable of monitoring (temporally) the 2004 and 2007 flood events. The analysis also provides an insight into the timing of streamflow, which explains why the events can be observed by GRACE. The final part of the paper also discusses the difference between ITSG and GFZ solutions.

I thank authors for sharing such a very interesting article. I find the paper very motivat-

ing particularly the use of a daily solution that has not really been exploited. I strongly recommend its consideration for publication.

The article is indeed well-written, but some clarification is still needed. I wish authors found my suggestions/comments below helpful for the revision:

1. Daily or monthly GRACE solutions? Authors generally use the term "GRACE", and it is quite confusing which solutions authors meant. For example, page 2, line 26 "Limited spatial sampling in East-West means that GRACE contributes . . .", I believe authors mean "daily GRACE" here. This is found throughout the text. Please consider using "daily GRACE" or "monthly GRACE" instead of "GRACE" where necessary.

2. ITSG or GFZ solutions? Similar to the above, please consider using "GRACE ITSG solution" or "GRACE GFZ solution" where necessary. For example, page 4, line 23 "The GRACE input is identical . . .", it is unclear what solution authors used here. Please consider including this information in Fig. 1 (a-c) as well.

Fig. 1, it is likely that only one solution is used for the analysis. Is it possible to present both solutions?

3. GRACE data processing. Can methodology presented in Eq. (1-4) be found in Kurtenbach et al. (2012)? If so, I think it can be removed from the paper since authors do not discuss the methodology further. Particularly the methodology regarding GFZ daily solution (which is more interesting) is not presented here at all. Could authors explain in details how GFZ daily solution is derived, or at least provide some references?

4. GRACE data processing. I believe the Sect. 2 can be organized better. I recommend using subsection here, like 2.1 ITSG, 2.1.1 daily ITSG, 2.1.2 monthly ITSG, 2.2 GFZ, 2.2.1 daily GFZ, 2.2.2 monthly ITSG, or 2.1 daily solution and 2.2 monthly solutions. The current format is shuffling around, which is quite confusing.

What are the maximum harmonic degrees of daily ITSG and GFZ solutions? 40?

5. GRACE data processing. Page 4, lines 6 – 17 are very confusing. Could you please

verify if the post-processing lines 6 – 12 is for the daily or monthly solutions? I guess it is "daily". Authors state that the GRACE Tellus procedure is used, but what about degree 1, destriping, smoothing and scaling (they are not mentioned in the paper)? It is likely that the spatial smoothing is not used (this is stated later in line 14), so I think referring to GRACE Tellus here is very misleading. Line 9, the daily GFZ uses degree 1 from SLR, but later on, authors cite Swenson et al. (2008). I am confused here which solution is really used. Also, are degree 1 and 2 available daily? Please consider rewriting this section.

For the monthly solution, why do authors use the DDK2 filter? This is quite aggressive filter compared to e.g., DDK3-DDK5, and might lead to significant attenuation of the TWSA amplitude. Authors discuss this issue later on page 5, lines 10 – 11.

What is the maximum degree used to computed TWSA in this study? Are they the same for both daily and monthly solutions?

6. The 2004 flood. Page 5, lines 4 – 5, why do the comparisons between daily ITSG and GFZ are based on a different day? If it does not require much work, it might be more informative to show all consecutive days of the flood periods instead of some particular snapshots. The visualization how flood distributes during the events will be very interesting. Line 13, please clarify what do you mean by "… while high TWSA values are slightly concentrated.". Line 15, "Flood stages were reached between 10 and 26 July …", this is not presented in Fig. 2 and 3, I think authors refer to the time series in Fig. 4. Please clarify.

7. Page 5, lines 10 – 11, this might not be fully due to the filter. The monthly value computed based on both flood and non-flood days likely show lower amplitude compared to the signal of the flood day.

8. Page 6, lines 1 – 2, authors present the correlation at specific epoch while the values in Table 1 is computed based on the entire time series. Is the correlation presented in line 1 – 2 significant? I suggest include all analyses into Table 1 and present Table 1

instead.

9. Please consider including the discussion of daily ITSG and GFZ performances in the conclusion. The daily ITSG tends to perform better in general and should be mentioned.

10. Page 7, Line 14, "seasonally-corrected LPF", is it daily or monthly?

11. Page 9, Line 2, the GFZ's ftp is not accessible. Is it possible to provide a public access address?
* * *

---

## Author Comment (AC1) · 10 Nov 2017

1. (P2,L14). Delivery of GRACE L1 takes 11 days as mentioned in introduction. How is it possible to produce the daily solution in 5 days? Do you have another improvement for the L1 data processing?

Authors' reply: The authors suggest to add the following sentence to the text in the manuscript (P2, L4): "Additionally, quick-look Level-1 data with a time delay of 1 day are made available on request by NASA Jet Propulsion Laboratory (JPL)."

2. (P3,Eq(3-4)). Please explain explicitly about k and N.

Authors' reply: The authors thank the reviewer for this observation and have phrased a reply also with reference to comment 3 of the second reviewer (see below). As the methodology of the derivation of the empirical covariance matrices is presented in detail in Kurtenbach et al., 2012, the equations in Section 2 have been removed from the text. We suggest to adapt the manuscript to as follows:

"Compared to monthly solutions, the limited spatial coverage within one day does not allow for GRACE to observe the full gravity field signal alone. Limited spatial sampling in East-West direction means that GRACE contributes little to no information to potential coefficients with orders higher than approximately 15. It is therefore necessary to introduce additional information to obtain reliable estimates of the full global gravity field signal. Applied to the determination of daily gravity field variations, this means that information on how the variable gravity field evolves with time is required. Since geophysical processes are not random, one can assume that the Earth's time-variable gravity field does not change arbitrarily from one day to the next. Kurtenbach et al., (2012) proposed to model this temporal evolution as a first-order Markov process, which can be fully described by its auto- and cross-covariance. Applied to daily GRACE solutions, the process to be modelled is the residual gravity field signal that is present in the observations after other effects, such as long-term, secular, as well as non-tidal ocean and atmosphere variations have been reduced. The main geophysical constituents left in the GRACE data are therefore continental hydrology, cryosphere, solid earth and errors in the background models (Kurtenbach et al., 2012). For the daily solutions of the ITSG-Grace2014 release (Mayer-Gürr et al., 2014) used in this analysis, the model output of the updated ESA Earth System Model (Dobslaw et al., 2015) is used to approximate the unknown covariance structure of this residual gravity field signal. The 6-hourly model output is resampled to one day using daily averaging. These daily averages are subsequently reduced by their sample mean, trend and annual signal. Finally, the empirical auto- and cross-covariance is computed from the resulting state vectors."

3. (3.1 Process dynamics). I understood that two daily solutions differ mostly from realization of matrix B that is empirically estimated by models. And, Figure 1 shows that solutions are close to each other regardless of models for B. Why don't you present the variance and co-variance matrix first?

Authors' reply: The gravity field solutions of GFZ and TUG do also differ significantly in how the GRACE data is processed. Figure 1 highlights the fact that the process dynamic itself does not greatly affect the solution – which means that the major part of the differences stem from the underlying GRACE processing strategies.

4. (P5,L4-9). Why the sampling dates of daily solutions are not the same? Is this due to noise so that some daily solutions could not present such nice snapshots? Indeed, GFZ RBF is corrupted more noise as shown in Figures 4 and 9.

Authors' reply: In part due to the different process dynamics of the daily gravity solutions causing different levels of noise, as pointed out by the reviewer, these peaks fall on different dates. This is -to some degree- indicated on the same page (P5, L30) with regard to the run off peak on 23 July 2004, which is reflected in HPF water storage signals of ITSG-Grace2014 and GFZ RBF on 21 and 24 July, respectively. Similarly, for the 2007 flood on P6, L21-22 it is offered: "As in 2004, the two daily solutions do not necessarily peak on the same day". The authors agree, however, the rational for the choice of sampling dates should be made more sufficiently clear. Therefore, the following line is suggested to be added to the text in the manuscript: "The sampling dates are chosen such that they represent the (local) peaks in the respective (high-pass filtered (HPF)) daily gravity time series corresponding with discharge peaks."

5. (P5, L9 "Due to the different. . ."). However, monthly mean of daily solutions need to be compared with monthly solutions. This is important because you examined the LPF of daily solutions. In terms of month-to-month variation, monthly solutions should be superior to monthly mean of daily solutions.

Authors' reply: The authors thank the reviewer for this suggestion, which we consider

correct. The focus of the paper, however, is on the added value of an increased temporal resolution of changes in the gravity field, i.e., the daily solutions, in terms of monitoring (extreme) hydrological events/floods. One could argue the increased capacity to monitor these events with a higher temporal resolution comes at the cost of signal accuracy, due to the Kalman filter.

6. (P5, L30) Correlation coefficients, r, in Table 1 are based on entire flooding years (am I right?). But you talked about r during flooding epochs in the text. I was quite confused during reading this part. So, please be clear this. It seems that many r in the table is not meaningful in the text. You may include r during flooding events in the table. Furthermore, r during flooding events are statistically significant? What are p values? (The 2007 flood) Similar comments for 'the 2004 flood' are also applied here.

Authors' reply: The authors thank the reviewer for pointing this out. The correlation coefficients (r) in Table 1 for the entire flooding years are contrasted with shorter (flooding) periods in the text, which exhibit higher correlation for shorter periods during which water storage peaks, as reflected in the LPF signal. Correlation coefficients during flooding are equally significant to those for the entire flooding year (p < 0.01), which are added to the text. To avoid confusion for the reader r for the entire flooding year and for the shorter flooding period is separated in Table and text in the revised manuscript.

7. (P7, L18) "..added value of the daily GRACE..") Did you mean ".. the monthly GRACE.."?

Authors' reply: In terms of short-term variations, the daily solutions do provide added value.

8. (Figure 10). Explanation for Figure 10 is quite terse. More explanations would be necessary for GRACE audience.

Authors' reply: Following the reviewer's comment, the text has been adjusted as follows: "This is further illustrated in Figure 10, which shows the frequency-separated

anomalies of river runoff vs. total water storage in the GBD for the flood years of 2004 and 2007 (Riegger and Tourian, 2014; Reager et al., 2015; Sproles et al., 2015). Generally, the hypothesis that storage can drive river runoff tends to indicate a slower process evolution through subsurface water storage and base flow generation, expressed by a strong correlation at longer (monthly) time scales. Daily HPF runoff is expected to be less correlated with and more variable than daily HPF storage, caused by precipitation than runs off quickly and doesn't enter storage for a significant amount of time. The fact that there is still a strong correlation between daily HPF storage and HPF river runoff, particularly for the 2007 flood (Table 1), points towards a scenario of increased storage in the river itself, in which the variation of daily total water storage reflects the inflow of river water into the delta. This water inflow can only be stored in the river during the time of flooding, when the available water storage in the area is at (near) capacity. However, while trends of the HPF signal of daily water storage and river runoff anomalies show agreement over periods of a few days, the higher frequency content of the daily solutions is not reflected in the daily river runoff. This high-frequency variation is attributed to process noise of the Kalman filter approach and a repeat period of 4 days for GRACE to pass over the GBD. Propagation of the full formal error matrix to the area mean value for each time step estimates the resulting noise level in the ITSG-Grace2014 and GFZ RBF daily solutions at approximately 1 cm TWSA and 1.45 cm TWSA, respectively. These numbers are confirmed by an empirical estimate using the methodology of Bonin et al. (2012). The difference in apparent noise in the two daily time series can primarily be attributed to the process models employed, which result in different temporal constraints and degree of spatial filtering."

9. (Figures 4 and 9) Both LPF time series quite differ from each other. This is not consistent with results in Figure 1.

Authors' reply: The gravity field solutions of GFZ and TUG also differ significantly in the way how the GRACE data is processed. Figure 1 highlights the fact that the process dynamics itself does not greatly affect the solution – which means that the major part

of the differences stem from the underlying GRACE processing strategies.

---

## Author Comment (AC2) · 10 Nov 2017

1. Daily or monthly GRACE solutions? Authors generally use the term "GRACE", and it is quite confusing which solutions authors meant. For example, page 2, line 26 "Limited spatial sampling in East-West means that GRACE contributes . . .", I believe authors mean "daily GRACE" here. This is found throughout the text. Please consider using "daily GRACE" or "monthly GRACE" instead of "GRACE" where necessary.

Authors' reply: The example on page 2, line 26 refers to the GRACE satellite(s). 'GRACE' is used in statements referring to the GRACE satellites and the GRACE satellite observations in general. 'daily' or 'monthly' is used throughout the manuscript if the

specific daily or monthly solutions are addressed, respectively.

2. ITSG or GFZ solutions? Similar to the above, please consider using "GRACE ITSG solution" or "GRACE GFZ solution" where necessary. For example, page 4, line 23 "The GRACE input is identical . . .", it is unclear what solution authors used here. Please consider including this information in Fig. 1 (a-c) as well. Fig. 1, it is likely that only one solution is used for the analysis. Is it possible to present both solutions?

Authors' reply: This analysis is indeed based on the input data for ITSG-Grace2014 only, we will add this information to the figure caption in the revised version of the manuscript. Our main goal here is to show that the prior information introduced through different process models has very little impact on the derived time series. Since the conclusion for the GFZ approach is the same, it is omitted here.

3. GRACE data processing. Can methodology presented in Eq. (1-4) be found in Kurtenbach et al. (2012)? If so, I think it can be removed from the paper since authors do not discuss the methodology further. Particularly the methodology regarding GFZ daily solution (which is more interesting) is not presented here at all. Could authors explain in details how GFZ daily solution is derived, or at least provide some references?

Authors' reply: The authors thank the reviewer for this observation. The methodology of the derivation of the empirical covariance matrices is indeed presented in detail in Kurtenbach et al. (2012). Also in line with a similar comment of Reviewer 1, we will adapt the manuscript to read: "Compared to monthly solutions, the limited spatial coverage within one day does not allow for GRACE to observe the full gravity field signal alone. Limited spatial sampling in East-West direction means that GRACE contributes little to no information to potential coefficients with orders higher than approximately 15. It is therefore necessary to introduce additional information to obtain reliable estimates of the full global gravity field signal. Applied to the determination of daily gravity field variations, this means that information on how the variable gravity field evolves with time is required. Since geophysical processes are not random, one can assume

that the Earth's time-variable gravity field does not change arbitrarily from one day to the next. Kurtenbach et al., (2012) proposed to model this temporal evolution as a first-order Markov process, which can be fully described by its auto- and cross-covariance. Applied to daily GRACE solutions, the process to be modelled is the residual gravity field signal that is present in the observations after other effects, such as long-term, secular, as well as non-tidal ocean and atmosphere variations have been reduced. The main geophysical constituents left in the GRACE data are therefore continental hydrology, cryosphere, solid earth and errors in the background models (Kurtenbach et al., 2012). For the daily solutions of the ITSG-Grace2014 release (Mayer-Gürr et al., 2014) used in this analysis, the model output of the updated ESA Earth System Model (Dobslaw et al., 2015) is used to approximate the unknown covariance structure of this residual gravity field signal. The 6-hourly model output is resampled to one day using daily averaging. These daily averages are subsequently reduced by their sample mean, trend and annual signal. Finally, the empirical auto- and cross-covariance is computed from the resulting state vectors."

Further, the following references are added detailing the GFZ daily solution processing:

Gruber et al., "Earth's time-variable GRACE gravity fields based on the RBF method evaluated by GPS, ICESat, hydrological modeling and altimetry satellite orbits." Submitted to Earth Surface Dynamics.

Gruber, "Short latency monitoring of continental, ocean- and atmospheric mass variations using GRACE inter-satellite accelerations", submitted to Geophysical Journal International.

4a. GRACE data processing. I believe the Sect. 2 can be organized better. I recommend using subsection here, like 2.1 ITSG, 2.1.1 daily ITSG, 2.1.2 monthly ITSG, 2.2 GFZ, 2.2.1 daily GFZ, 2.2.2 monthly ITSG, or 2.1 daily solution and 2.2 monthly solutions. The current format is shuffling around, which is quite confusing.

Authors' reply: The authors thank the reviewer for pointing this out. We will restructure

the manuscript the following way:

2.1 Computation of Daily GRACE Solutions -ITSG -GFZ

2.2 Derivation of Gridded Water Storage Anomalies from Daily and Monthly GRACE solutions -ITSG (daily/monthly) -GFZ (daily/monthly)

This should also alleviate to structure the post-process strategies (Reviewer comment number 5) in a clearer manner.

4b.What are the maximum harmonic degrees of daily ITSG and GFZ solutions? 40?

Authors' reply: The ITSG-Grace2014 daily solutions are published up to a maximum degree of 40. Since the GFZ daily solutions are parametrized in space domain, no exact maximum degree can be given here. However, they exhibit a similar spatial resolution as the ITSG-Grace2014 solutions.

5. GRACE data processing. Page 4, lines 6 – 17 are very confusing. Could you please verify if the post-processing lines 6 – 12 is for the daily or monthly solutions? I guess it is "daily". Authors state that the GRACE Tellus procedure is used, but what about degree 1, destriping, smoothing and scaling (they are not mentioned in the paper)?It is likely that the spatial smoothing is not used (this is stated later in line 14), so I think referring to GRACE Tellus here is very misleading. Line 9, the daily GFZ uses degree 1 from SLR, but later on, authors cite Swenson et al. (2008). I am confused here which solution is really used. Also, are degree 1 and 2 available daily? Please consider rewriting this section.

Authors' reply: As stated in the reply to reviewer comment number 4, a dedicated section for the derivation of gridded water storage anomalies will be introduced and the content will be restructured. The manuscript will then read:

"2.2 Derivation of Gridded Water Storage Anomalies from Daily and Monthly GRACE solutions To derive gridded TWSA from daily and monthly gravity field potential coefficients, the general processing scheme used for the GRACE land water mass grids

provided by GRACE Tellus is followed (Swenson, 2012; Landerer and Swenson, 2012). This post-processing scheme can be split into three steps: 1) replace the C20 coefficient (Earth's oblateness), 2) transform the potential coefficients into center of Earth by adding degree one coefficients and 3) apply a spatial filter to the coefficients. For the ITSG-Grace2014 daily solution, the C20 coefficient is replaced by linear interpolation of the monthly satellite laser ranging time series provided by Cheng et al. (2011). Similarly, daily degree one coefficients are obtained by linear interpolation of the monthly time series provided through Tellus, based on the methodology described in Swenson et al. (2008). Glacial isostatic adjustment (GIA) correction has been applied using the model from A et al. (2013). Since the daily GRACE solutions are constraint within the least squares adjustment, no additional spatial filtering is necessary. For comparison, the monthly solutions of ITSG-Grace2014 (Mayer Gürr et al., 2014) are also considered. Here, the same models as for the daily solutions are applied. However, since the monthly solutions are unconstraint, the coefficients were smoothed by a DDK2 anisotropic filter (Kusche et al., 2009). The GFZ daily solutions make use of the same models as are used in the post-processing of ITSG-Grace2014, except for degree one, which is also taken from a SLR estimate (Cheng et al. 2010). As with the daily ITSG solution, no additional spatial filtering was performed. The monthly solutions from GFZ (RL05a, Dahle et al., 2012) are corrected using the same models and DDK2 filter is applied. Both daily and monthly solutions are then propagated to TWSA on a 1âĄř x 1âĄř grid (∼100 km at 25âĄř latitude). The actual spatial resolution of the gridded TWSA, however, is lower with approximately 330 km and 500 km for monthly and daily solution, respectively."

6. For the monthly solution, why do authors use the DDK2 filter? This is quite aggressive filter compared to e.g., DDK3-DDK5, and might lead to significant attenuation of the TWSA amplitude. Authors discuss this issue later on page 5, lines 10 – 11.

Authors' reply: Three filters were applied (DDK1-3) to the monthly solutions, of which only one – the middle choice in terms of rigor - for reasons of conciseness was further

explored for comparison to the daily gravity solutions. The authors like to point out that rather than the (difference in) amplitude, the dynamic of the two temporal solutions is relevant here. Additionally, the spatial averaging of the gridded data to basin averages may have an attenuation effect of the TWSA amplitude comparable to the DDK filter(s).

7. What is the maximum degree used to computed TWSA in this study? Are they the same for both daily and monthly solutions?

Authors' reply: Both daily and monthly solutions are used up to the maximum degree (40 for daily and 90 for monthly). Due to the spatial low pass filtering, the remaining signal contents is however considerably lower than degree 90.

8. The 2004 flood. Page 5, lines 4 – 5, why do the comparisons between daily ITSG and GFZ are based on a different day? If it does not require much work, it might be more informative to show all consecutive days of the flood periods instead of some particular snapshots. The visualization how flood distributes during the events will be very interesting.

Authors' reply: The authors refer to the reply to comment 4, Reviewer 1). As the TWSA maps do not show very discernable changes from day-to-day, the visualization is chosen such it synoptically shows the largest changes before, at and after peak flow.

9. Line 13, please clarify what do you mean by ". . . while high TWSA values are slightly concentrated.".

Authors' reply: 'Concentrated' means 'focused', in that the maximum values cover a smaller geographic area. The authors propose to replace 'concentrated' with 'focused' in the text to clarify.

10 Line 15, "Flood stages were reached between 10 and 26 July . . .", this is not presented in Fig. 2 and 3, I think authors refer to the time series in Fig. 4. Please clarify.

Authors' reply: The flood stages are partly presented in Fig. 2 and 3, in the sense that

the chosen dates fall within the indicated period. On the other hand, the authors agree that the sentence also refers to Fig. 4. In the text no reference is made to any of the figures, but rather to independent information from the literature (Hopson and Webster, 2010).

11. Page 5, lines 10 – 11, this might not be fully due to the filter. The monthly value computed based on both flood and non-flood days likely show lower amplitude compared to the signal of the flood day.

Authors' reply: The authors agree with the reviewer. The text in the manuscript is adapted to include the reviewer's suggestion: "Due to, amongst others, temporal averaging and the different filtering techniques (see above), the signal amplitude of the daily and monthly gravity field solutions also cannot be compared directly."

12. Page 6, lines 1 – 2, authors present the correlation at specific epoch while the values in Table 1 is computed based on the entire time series. Is the correlation presented in line 1 – 2 significant? I suggest include all analyses into Table 1 and present Table 1 instead.

Authors' reply: The authors thank the reviewer for this suggestion. P values (<0.01) are added to the revised version of the manuscript. In order to avoid confusion, Table 1 lists the correlation coefficients of the entire flood period only and the specific epoch are highlighted in the text.

13. Please consider including the discussion of daily ITSG and GFZ performances in the conclusion. The daily ITSG tends to perform better in general and should be mentioned.

Authors' reply: The authors suggest to add the following text:

"While the ITSG-Grace2016 shows relatively higher correlation with river flow and higher temporal consistency, the GFZ RBF solution exhibit a better spatial focus of the flooded area, possibly indicating a higher content of the hydrological signal."

14. Page 7, Line 14, "seasonally-corrected LPF", is it daily or monthly?

Authors' reply: HPF/LPF analysis has only been applied to the daily series, so it is daily.

15. Page 9, Line 2, the GFZ's ftp is not accessible. Is it possible to provide a public access address?

Authors' reply: The GFZ-RBF solutions are made accessible as total water storage grids in equivalent water height via ftp://gfzop.gfz-potsdam.de/EGSIEM/ and will be made public via ICGEM as spherical harmonic data sets, as well.